# DynamicRTL: RTL Representation Learning for Dynamic Circuit Behavior

## Abstract

There is a growing body of work on using Graph Neural Networks (GNNs) to learn representations of circuits, focusing primarily on their static characteristics. However, these models fail to capture critical runtime behavior, which is crucial for tasks like hardware verification and optimization. To address this limitation, we introduce DynamicRTL, a novel GNN-based approach that learns circuit representations by incorporating both static structures and multi-cycle execution behaviors. DynamicRTL leverages an operation-level Control Data Flow Graph (CDFG) to represent Register Transfer Level (RTL) circuits, enabling the model to capture dynamic dependencies and runtime execution. To train and evaluate DynamicRTL, we built the first comprehensive dynamic circuit dataset, comprising over 6,300 Verilog modules and 190,000 simulation traces. Our results demonstrate that DynamicRTL consistently outperforms existing models in branch prediction tasks. Furthermore, its learned representations transfer effectively to related tasks, achieving strong performance in assertion prediction and underscoring its transfer learning capabilities for dynamic circuit tasks.

## 1 Introduction

Over the past decades, Moore's law has driven the exponential growth in the scale of digital circuits. However, as the circuit complexity increases, traditional algorithms and methodologies for circuit design and optimization are increasingly challenged to meet the pressing demands for time and cost efficiency. There is an urgent need for new models, representations, and methodologies for circuit understanding to advance research in hardware design and electronic design automation (EDA), which are essential for empowering engineers to create high-performance hardware solutions.

Recent works have begun to solve many canonical tasks in hardware development with deep learning methods (Chen et al., 2024; Ma et al., 2020). By training on extensive circuit data, models have demonstrated the potential to understand the static feature of circuits and outperform traditional methods on some prediction tasks, including circuit quality prediction (performance, power, area) (Sengupta et al., 2022; Fang et al., 2023; Lopera et al., 2021), combinational functionality prediction (Wang et al., 2022; Li et al., 2022; Shi et al., 2023) and so on.

Despite these notable achievements, it has been observed that these models struggle with tasks that require in-depth analysis of circuit designs, especially those involving the dynamic behavior of circuits such as hardware verification, dynamic power estimation and so on (Khan et al., 2024; Vasudevan et al., 2021). The main reason is that current models exclusively depend on the static data of circuits (*i.e.,* hardware source code, netlist) as input. As a result, these models are limited to learning only structural or semantic information about the circuits.

However, the dynamic behavior of circuits, which involves changes in circuit states over multiple operational cycles, is equally important for understanding circuits and facilitating dynamic-related downstream tasks. These behaviors can reveal complex dependencies and interactions that are unapparent in static representation, thereby significantly enhancing the quality of circuit representation.

In this study, we diverge from traditional approaches learning static representations of circuits. In an innovative endeavor, we aim to train the model to learn the dynamic representation of circuits based on their multi-cycle execution behavior. We focus on two dynamic tasks: branch prediction

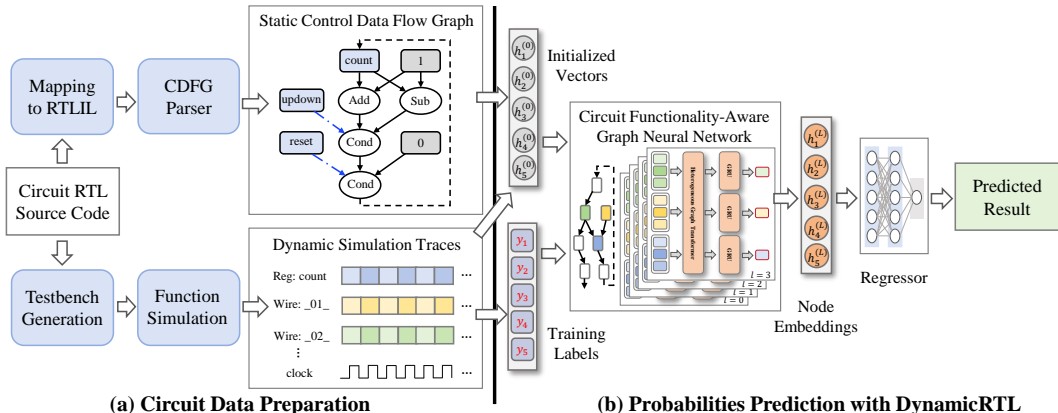

**(a) Circuit Data Preparation**          **(b) Probabilities Prediction with DynamicRTL**

Figure 1: The overview of DynamicRTL.

and assertion prediction. These tasks are essential in hardware verification and can facilitate early evaluation of test cases during simulation-based verification processes (Vasudevan et al., 2021).

We design the model DR-GNN (**D**ynamic**R**TL-GNN) to learn the dynamic behavior of circuits. Our model is distinguished by two aspects. First, we construct the graph neural network (GNN) based on the operation-level Control Data Flow Graph (CDFG) of Register Transfer Level (RTL) circuit design. This CDFG more accurately reflects the functionality of circuit, thereby facilitating the learning of dynamic behaviors. Second, we design the circuit functionality-aware GNN propagation mechanism that integrates specific circuit features into the model. We incorporate information about node operation types, positions, and dynamic embeddings into a heterogeneous graph transformer (Hu et al., 2020) to capture both the structural and dynamic characteristics of circuits.

We leverages a comprehensive collection of Verilog designs as our dataset. Because the dynamic behavior of circuits can be fully encapsulated by the execution of its branches, we utilize branch execution information as our training supervision. DynamicRTL has exhibited outstanding performance in the task of branch prediction, significantly surpassing the results of models that rely solely on structure and semantic learning. Moreover, DynamicRTL can also generate useful representations of circuits for indirectly related downstream tasks, and we demonstrate the transfer learning on assertion prediction problem. To our knowledge, DynamicRTL is the first unified model that can capture the dynamic behavior of various hardware designs, setting the stage for teaching neural network models to better understand how circuits execute.

In summary, this paper makes the following contributions:

- We introduce the problem of learning circuit dynamic representation. We create the first dynamic circuit dataset, comprising over 6,300 diverse Verilog designs, along with their corresponding CDFGs and 190,000 simulation traces.

- We design the DR-GNN model which is built on the circuit operation-level CDFG and is aware of the dynamic functionality of circuits.

- We get the first unified representation for circuit dynamic behaviors and achieve the state-of-the-art performance in circuit branch prediction with an accuracy of 93.48%.

- We show that DR-GNN representations pre-trained on branch prediction are useful for transfer learning, achieving an average accuracy of 88.05% on assertion prediction task.

## 2   PRELIMINARY

Figure 1 presents an overview of DynamicRTL. In this section, we discuss several key concepts related in our workflow. We introduce our method for representing circuit source code as a graph structure. Then, we explain the concept of dynamic behavior in circuits. Finally, we outline the tools used to extract these information from circuit designs.

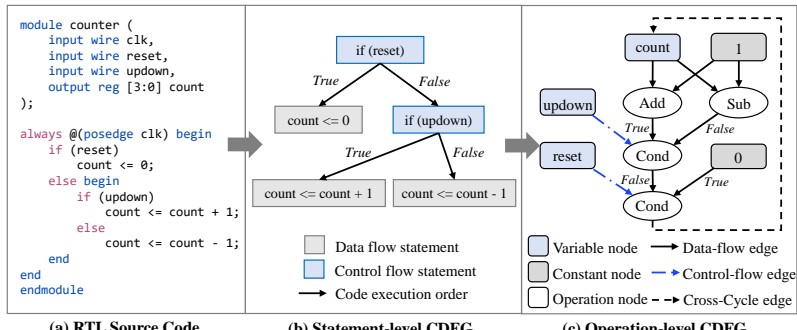

**(a) RTL Source Code**  **(b) Statement-level CDFG**  **(c) Operation-level CDFG**

Figure 2: Comparison between statement-level CDFG and operation-level CDFG. In Verilog grammar, the operator <= denotes a non-blocking assignment to register.

**RTL Code.** *Register transfer level* (RTL) is a design abstraction level used in digital circuit design. A digital circuit usually composed of combinational logic (computing operators, branch controls) and sequential logic (registers). The assignment in RTL represents a physical connection between wires or registers. In this context, a wire is a conductive path for signal transmission, while a register is a small storage element that holds data temporarily during the execution of a digital circuit. RTL is a representation that focuses on the flow of data between registers and the operations performed on that data. The register transferring data flows are commonly written in `always` blocks, meaning that the logic is executed during every clock cycle. This is analogous to a software program written in `while` loop, where the same code is repeatedly executed and the state continuously changes.

**Control Data Flow Graph.** A common way of understanding the function of RTL is *control data flow graph* (CDFG). Figure 2 shows an example of RTL code and the built CDFGs.

Previous work, Design2Vec (Vasudevan et al., 2021), uses a statement-level CDFG to learn semantic representation of circuits, where each node represents an RTL statement. However, this CDFG only reflect the execution of circuit code from a software perspective, which fails to reflect the actual data flow within circuits. For example, in Figure 2 (b), three distinct statements assign different values to the register `count` under varying conditions. This representation diverges from the true dynamic behavior of the circuit. Actually, circuits operate in parallel, processing multiple signals simultaneously. Their behavior is determined by the complex interconnections and timing of components.

To more accurately represent the dynamic behavior of circuits, we propose the operation-level CDFG as the graph structure to be used by our model. In our CDFG, directed edges indicate data or control flows between nodes. In sequential circuits, registers store values to the next clock cycle, therefore the in-edges of registers signify data flow across clock cycles, making the CDFG a cyclic graph. The CDFGs comprise three node types: variable nodes, constant nodes and operation nodes.

*Variable nodes* represent variables that have dynamic values, such as wires and registers, which can only be assigned once in RTL and have a single in-edge in CDFG. Input wires are special variable nodes whose value depends on the external drive and have only out-edges without in-edges. *Constant nodes* represent unchanging values in a circuit. These nodes also have no in-edges and represent the same values throughout clock cycles.

*Operation nodes* represent operations in data flow, with one or more in-edges and a single out-edge. The `condition` node is a unique type of operation node, functioning like a multiplexer, and uses a one-bit select signal to control data flow from different data channels. Since some operators do not satisfy the commutative law, the order of in-edges must not be overlooked.

The nodes in CDFG can be multi-bit, allowing for a higher level of abstraction and a more effective way to represent RTL dynamic behaviors compared with single-bit circuit graphs (Fang et al., 2023; Li et al., 2022), which are closer to netlists. For more details on circuit CDFG, refer to Appendix B.

**Circuit Dynamic Execution State.** The dynamic state of a circuit refers to the set of values that change as the circuit executes. Input signals drive the internal wires based on the combinational operations. Some values are stored in registers to be used in the next clock cycle. In principle, the dynamic state of a circuit is represented by the value sequence of its registers. However, to

achieve a comprehensive representation of the circuit, it is essential to consider both sequential and combinational features. As a result, we also take the values of wires into account when assessing dynamic states. These values can be easily obtained through hardware simulation.

**Tools.** To simplify the process of constructing the CDFG, we first use the Verilog synthesis tool Yosys (Wolf et al., 2013) to convert the source RTL code into an *RTL intermediate language* (RTLIL). This conversion results in a functionally equivalent description, consisting only of assign statements and register transferring. Next, we use the Stagira Verilog parser (Chen et al., 2023) to parse the RTLIL and generate an *abstract syntax tree* (AST). Finally, we traverse the AST to create the circuit's CDFG. To collect the dynamic traces of the circuits, we use Verilator (Snyder et al., 2023) as simulator and capture the value of wires and registers in each clock cycle.

## 3 CIRCUIT DYNAMIC TASKS

Our model is designed to focus on two tasks that are strongly related with circuit dynamic behaviors, branch prediction and assertion prediction. Given an input sequence and a design, the model predicts whether a specific branch can be hit or a particular assertion can be satisfied.

Hardware branches, often represented by conditional statements like `if` or `case` in RTL code, denote points where execution can follow multiple paths based on certain conditions. These branches are important coverage metrics in hardware verification, ensuring each code branch is executed at least once to gauge testing thoroughness. In the transformation to CDFG, branches are mapped to `condition` nodes, with the select signals controlling the selection of data flow.

Hardware assertion is used to validate that specific conditions hold true under certain circumstances. It provides a framework for writing constraints, checkers and cover points for hardware designs. For example, in an arbiter whose function is to arbitrate between two request ports (`A` and `B`), we can assert `!(grant_A && grant_B)` to ensure at most one request is granted access at the same time.

In our study, we use branch prediction as supervision, which will be detailed in Section 4.4. Branch prediction provides a consistent task across different designs and the supervision is easy to obtain. Meanwhile, the execution of a branch encapsulates most of the dynamic behavior in circuits. If the model accurately predicts branch hitting, it implies that the model has effectively learned the dynamic information of other wires and registers governing the execution of these branches. After training on branch information, to further investigate whether the learned representation contains comprehensive dynamic circuit behaviors, we use assertion prediction as a downstream task, which will be discussed in detail in Section 5.4.

## 4 DR-GNN: CIRCUIT FUNCTIONALITY-AWARE GNN MODEL

### 4.1 OVERVIEW

Our model takes as input the CDFG and input sequences $I$. The circuit has branch set $\mathbb{B}$. The main objective of our model is to predict the probability $is\_hit(I, \mathbb{B})$ that input $I$ hit branches in $\mathbb{B}$.

Figure 3 presents the architecture of DR-GNN. The model operates in three stages: Firstly, the initial embedding for each circuit node is assigned based on its attribute and functionality (Section 4.2). Secondly, the DR-GNN updates the node embeddings using a circuit functionality-aware propagation method that accommodates the behaviors of circuit operations (see Section 4.3). Thirdly, after the final embeddings for each node are obtained, they are utilized to compute the loss for various prediction tasks related to circuit dynamic behaviors (Section 4.4).

### 4.2 CIRCUIT NODE INITIALIZATION

**Input Sequence Embedding.** The dynamic behavior of a circuit is intrinsically related to its input. Therefore, we begin by embedding the input sequence as the initial embedding of the input node. This process involves two steps: embedding the value in each clock cycle, and then integrating these value embeddings into a sequence embedding.

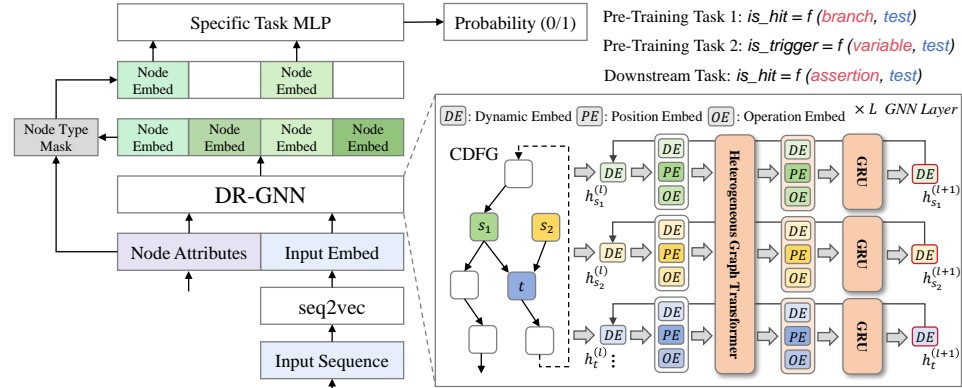

Figure 3: The overview of circuit functionality-aware DR-GNN model.

Because many circuit operations function at the bit-level, we treat the value in each clock cycle as a binary vector and embed it with linear projection, similar to previous work on numerical representation (Yan et al., 2020). This is equivalent to defining a learnable vector for each bit position, and summing these vectors element-wise, modulated by the value of corresponding bit. For the learnable vector $e_i$ for each bit and an $n$ bit value $v = \{b_1, b_2, ..., b_n\}$, we compute the vector embedding $v_{emb} = \sum_{i=1}^{n} b_i e_i$. Subsequently, we use the Gated Recurrent Unit (GRU) to embed the value in each clock cycle into a sequence embedding and use that as the embedding of the input node.

**Constant Value Embedding.** In order to ensure the embedding consistency of each node and reduce the complexity of GNN learning, we employ a strategy similar to input node embedding for embedding the constant node. The constant node can be treated as an input node that keeps the same value throughout each clock cycle. We assign that an initial embedding similar with the input node.

**Operation Node Embedding.** The embedding of operation nodes represents the dynamic behavior of their calculated results. Since we cannot predict these data before simulation, we initialize this embedding to all zeros, indicating the unknown. A particular instance here is the register node. In sequential circuits, registers do not perform calculation and only store values to the next clock cycle. For the dynamic behavior of circuits, the register functions as an operator in fact. So we also treat the register as an operation in the model.

### 4.3 CIRCUIT FUNCTIONALITY-AWARE PROPAGATION

In hardware circuit, data is processed through various hardware components, which correspond to the operation nodes in our CDFG. We can use the aggregation mechanism in GNN to represent this dynamic behavior. However, there are challenges due to the different types of operations in circuit logic. Each operation requires a specific aggregation pattern to properly handle the embeddings of neighboring nodes. Additionally, some operators do not satisfy the commutative law. For instance, the subtraction operator yields different results for `a-b` and `b-a`. Therefore, it is also essential to consider the position of source nodes when aggregating embeddings to the target node.

To solve this problem, we design a novel aggregation method in DR-GNN that effectively represents circuit logic. Because the circuit CDFG is a heterogeneous directed graph, we build the aggregation mechanism referring to the Heterogeneous Graph Transformer (HGT) (Hu et al., 2020). We use different operation types and node positions to parameterize weight for heterogeneous attention, thereby achieves effective aggregation for different circuit operations.

Consider an operation node $t$ which has embedding $h_t^{(l)}$ in the $l$-th GNN layer for its dynamic circuit behavior. The node $t$ has multiple source nodes $s_1, s_2, ..., s_m$, each with their respective dynamic embeddings $h_{s_1}^{(l)}, h_{s_2}^{(l)}, ..., h_{s_m}^{(l)}$. The operation node type is denoted as $op[t]$. The position of source node is denoted as $pos[s, t]$, which reflect the order of source node $s$ in the context of operation $t$. We concatenate the operation type embedding and the source node position embedding with the dynamic embedding of source node to be the input of heterogeneous graph transformer.

$$x_s^{(l)} = \text{Concat}(h_s^{(l)}, \text{Embed}(op[t]), \text{Embed}(pos[s, t]))$$

In heterogeneous graph transformer, we map the concatenated embedding of source node $s$ and the embedding of target node $t$ into Query and Key vectors. We calculate the concatenation of these vectors as attention. The attention weight of node $s$ to $t$ is calculated by

$$w_{s,t}^{(l)} = \text{MLP}_{\text{attn}}(\text{Concat}(\text{Q-Linear}(\boldsymbol{x}_s^{(l)}), \text{K-Linear}(\boldsymbol{h}_t^{(l)})))$$

For each source node $s$ connected to node $t$, we calculated its normalized attention weight by

$$\{a_{s_1,t}^{(l)}, a_{s_2,t}^{(l)}, ..., a_{s_m,t}^{(l)}\} = \text{Softmax}(\{w_{s_1,t}^{(l)}, w_{s_2,t}^{(l)}, ..., w_{s_m,t}^{(l)}\})$$

We can get the message from source node $s$ to target node $t$ by a Value vector.

$$message_{s,t}^{(l)} = \text{V-Linear}(\boldsymbol{x}_s^{(l)})$$

We merge messages from source nodes with attentions and get aggregation information for node $t$.

$$aggr_t^{(l)} = \sum\nolimits_{i=1}^{m} message_{s_i,t}^{(l)} \cdot a_{s_i,t}^{(l)}$$

Finally, we use GRU to update the embedding of target node $t$, where $aggr_t^{(l)}$ is the aggregation information as the GRU input and $\boldsymbol{h}_t^{(l)}$ is the past state of GRU. The output $\boldsymbol{h}_t^{(l+1)}$ will serve as the dynamic embedding for the subsequent $(l + 1)$-th GNN layer.

$$\boldsymbol{h}_t^{l+1} = \text{GRU}(aggr_t^{(l)}, \boldsymbol{h}_t^{(l)})$$

The use of GRU is essential in this context, because during each layer of GNN propagation, nodes aggregate information from neighboring nodes that are one step further. To ensure that each node achieves an overall receptive field across the circuit, the number of GNN layers must match or exceed the depth of the circuit's CDFG. Relying solely on basic node update methods can lead to over-smoothing of GNN, so we use GRU to solve this issue.

Finally, by stacking $L$ layers, we obtain the node representations of the entire graph, denoted as $\boldsymbol{h}^{(L)}$. These representations can be used for end-to-end training or fed into downstream tasks.

### 4.4 PRE-TRAINING TASK

We train the model with two tasks. Task 1 involves predicting the branch hit probability. We identify the nodes serving as select signals of the `condition` node and form the branch set $\mathbb{B}$. We readout the embedding $h_b^{(L)}$ of node $b$ and pass it through a multi-layer perceptron to get the probability.

$$\hat{P}_b = \text{MLP}_{\text{branch}}(\boldsymbol{h}_b^{(L)}), \quad b \in \mathbb{B}$$

Task 2 involves predicting the variable trigger probability. We select the variable nodes not in branch set $\mathbb{B}$ and form the variable set $\mathbb{V}$. A variable is considered triggered if its value signal has been assigned at least once during execution. We use the embedding $h_v^{(L)}$ of node $v$ to get the probability.

$$\hat{P}_v = \text{MLP}_{\text{variable}}(\boldsymbol{h}_v^{(L)}), \quad v \in \mathbb{V}$$

Actually, the value trigger information is incorporated in the branch hit information. When a branch is hit, the assignment inside the branch is executed, and the corresponding variable is triggered. The reason for introducing this additional supervision is to guide the model in focusing more on the fine-grained aspects of variable behaviors, rather than just the high-level branch behavior. In Section 5.4, we will demonstrate how this additional supervision significantly aids in transferring the model to the downstream task of assertion prediction.

## 5 EXPERIMENTS

### 5.1 DATASET PREPARATION

To train the DynamicRTL, we build the first dynamic circuit dataset, which consists of around 6,300 different circuit designs and 190,000 circuit simulation traces.

**Circuit Designs Collection.** We begin by collecting some existing Verilog datasets including MG-Verilog (Zhang et al., 2024) and VeriGen (Thakur et al., 2024). To further expand our collection, we searched GitHub using keywords such as Verilog, RTL, circuit and subsequently scraped Verilog code files from the relevant repositories. We remove designs with syntax error and those that failed during translation to CDFG or simulation. Besides, we only keep sequential circuits in our dataset. As a result, our dataset comprises approximately 6,300 usable Verilog modules. The circuit sizes of these designs range from 10 to over 500 CDFG nodes, with an average of 51 nodes per design. This corresponds to an RTL design containing more than 100 lines of Verilog RTL code. It should be noted that the primary goal of constructing a world-level CDFG for circuits is to achieve a compact representation. Consequently, while the scale of our CDFGs is relatively small, they can effectively represent circuits of significantly larger scale. For instance, an RTL-level circuit graph with 100 nodes can be transformed into an AIG graph with over 10,000 nodes after synthesis, which results in the loss of Verilog's semantic information. The collected circuits correspond to Verilog files with up to more than 1,000 lines and netlists with up to 50,000 nodes. Further information about our circuit dataset is available in Appendix C.

**Simulation Traces Collection.** We use Verilator (Snyder et al., 2023) to simulate collected Verilog modules. We generate random input patterns and simulate each module with 30 different traces. During simulation, values of all the internal variables are collected in each clock cycle. When generating testbench, we automatically identify special signals such as the reset signal. We make that the reset signal is active only at the beginning of simulation and remains inactive thereafter. This approach guarantees the simulation traces encompass a comprehensive range of circuit behaviors.

## 5.2 EXPERIMENTAL SETUP

Our dataset consists of 6,300 unique circuit CDFGs, corresponding to Verilog files with over 1,000 lines and netlists containing more than 50,000 nodes. Each CDFG is paired with 30 simulation traces, resulting in approximately 190,000 data entries. We split our dataset by design, using 80% of the designs and their traces for training and 20% for evaluation. This approach ensures that the designs used for evaluation are never seen by the model during training, providing a reliable measure of its generalization ability. The binary cross-entropy losses of the pretraining tasks are calculated and summed to form the final loss. The hyperparameters of our model are detailed in Appendix A.2. We train our model on a single A800 GPU for 60 epochs, using a learning rate of 0.0001 and the Adam optimizer.

## 5.3 PRE-TRAINING TASK

We evaluate the DR-GNN model on the task of branch prediction and variable trigger prediction. We compare our model with several variants of graph neural network: graph convolution network (GCN) (Kipf & Welling, 2016), graph attention network (GAT) (Veličković et al., 2017), and Design2Vec (Vasudevan et al., 2021). For both GCN and GAT, we initialize the embedding of input and constant nodes to be the same as DynamicRTL. The only exception is that we do not use the circuit functionality-aware propagation methods. We set the node embedding of operation nodes with their attribute embeddings. These embeddings are then propagated using the mechanisms of GCN and GAT. For Design2Vec which constructs the GNN based on statement-level CDFG, we reproduce its approach by initializing each node's embedding with the semantic embedding of the corresponding statement. We use GCN as the propagation model in Design2Vec and concatenate the final embedding of each statement with the embedding of the input sequence, which is subsequently passed through a linear layer to predict the hit probability of the statement.

Table 1 presents the experiment results. The DR-GNN model outperforms GCN, GAT, and Design2Vec&GCN, achieving branch prediction accuracy of approximately 93% across input sequence lengths of 10, 20, and 30, as well as the variable trigger prediction accuracy of about 88%. Different from other models, which exhibit a decline in performance as the sequence length increases, our model maintains robust performance for longer input sequence. The experiment results demonstrate the effectiveness of our circuit functionality-aware propagation method and illustrate the advantages of learning circuit dynamic behaviors on the operation-level circuit CDFG.

Besides, we evaluate the performance of our model on circuit designs of different size, as presented in Table 2. For designs with fewer than 50 nodes, our model achieves over 95% branch hit accuracy

Table 1: Comparison of different models for branch prediction and variable trigger prediction tasks.

| | Branch Hit Prediction | | | Variable Trigger Prediction | | |
|---|---|---|---|---|---|---|
| Sequence length | 10 | 20 | 30 | 10 | 20 | 30 |
| GCN | 80.38 | 84.65 | 80.71 | 65.47 | 69.44 | 66.06 |
| GAT | 82.27 | 80.67 | 78.67 | 67.75 | 67.62 | 66.91 |
| Design2Vec&GCN | 81.24 | 74.14 | 69.15 | 69.68 | 60.37 | 62.64 |
| DR-GNN w/o PE | 91.65 | 90.96 | 90.9 | 81.25 | 82.67 | 84.82 |
| DR-GNN w/o OP | 86.79 | 91.3 | 88.69 | 81.03 | 81.42 | 82.61 |
| DR-GNN w/o GRU | 87.94 | 88.27 | 87.14 | 74.28 | 76.79 | 75.09 |
| **DR-GNN** | **93.16** | **93.48** | **92.25** | **88.29** | **89.13** | **87.17** |
| Statistical frequency | 59.57 | 62.14 | 64.51 | 56.33 | 60.53 | 61.37 |

Table 2: Model performance on circuit designs of different sizes.

| Node Number | Branch Hit | Variable Trigger |
|---|---|---|
| [10, 30) | 97.22 | 93.35 |
| [30, 50) | 95.20 | 92.45 |
| [50, 100) | 93.12 | 90.07 |
| [100, 200) | 92.46 | 89.64 |
| [200, 500] | 90.52 | 84.94 |

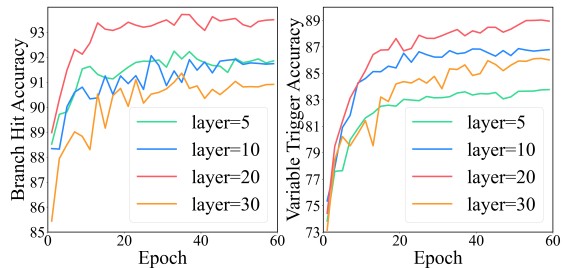

Figure 4: Effect of GNN layer number on DR-GNN performance.

and more than 93% variable trigger accuracy. As the design size increases, understanding becomes more challenging, while our model maintains good performance. Additionally, we investigate the impact of the number of GNN layers in our model. The results in Figure 4 indicate that DR-GNN with 20 layers yields the best performance. In our circuit dataset, the average depth of circuit CDFG is 7. If the number of layers is less than this depth, the node's receptive field fails to encompass overall circuit. Conversely, using a GNN with too many layers can lead to over-smoothing issue. DR-GNN with 20 layers strikes an optimal balance between these two concerns.

We also evaluate the DR-GNN model through ablation experiments. In DR-GNN, we integrate the embeddings of node position and operation type as part of the input of heterogeneous graph transformer to make our model circuit functionality-aware. Additionally, we employ a GRU for node update to mitigate GNN over-smoothing. Table 1 presents the experiment results comparing the original DR-GNN model with its ablated versions: DR-GNN without position encoding (DR-GNN w/o PE), DR-GNN without operation type encoding (DR-GNN w/o OP), and DR-GNN without GRU node update (DR-GNN w/o GRU). The results indicate that the original DR-GNN model outperforms its variants, which demonstrates the effectiveness of each component in our circuit functionality-aware DR-GNN model and corroborates our analysis in Section 4.3.

## 5.4 DOWNSTREAM TASK

We use the representation learned by DR-GNN for assertion prediction. This downstream task also help us understand what dynamic behaviors that the model has learned. The representation of each circuit CDFG node is pre-trained in Section 5.3 with the input sequence length of 20. For this experiment, we trained an extra multi-layer perceptron for each assertion. The model outputs the prediction whether a variable satisfies the assertion under specific input. We totally write 8 assertions. For single-variable assertions, we select every possible variable in circuit as prediction target. For dual-variable assertions, we target possible variable pairs.

Table 3 presents the assertions and their prediction accuracies with the DR-GNN representations. Our results indicate that the learned embeddings effectively predict some simple assertions. The

Table 3: Assertion prediction accuracy with DR-GNN circuit representations.

| Assertion | DR-GNN$_{vs}$ | DR-GNN$_{bs}$ | **DR-GNN** | Statistical frequency |
|---|---|---|---|---|
| `v < 4` | 81.90 | 81.34 | **86.81** | 57.02 |
| `v < 16` | 81.32 | 82.24 | **87.02** | 61.67 |
| `v ≠ 2` | 80.70 | 79.23 | **81.80** | 71.95 |
| `v ≠ 4` | 80.16 | 76.22 | **81.03** | 74.87 |
| `v1 ≠ v2` | 88.31 | 84.51 | **91.04** | 68.91 |
| `v1 < v2` | 81.51 | 76.88 | **88.28** | 50.74 |
| `v1 & v2 = 0` | 82.81 | 66.48 | **86.79** | 61.62 |
| `v1 \| v2 ≠ 0` | 89.07 | 85.28 | **91.65** | 69.83 |

model accurately assesses the approximate value ranges for assertions like `v<4` and `v<16`. This capability is also evident in assertions requiring variable comparisons, such as assertion `v1≠v2` and `v1<v2`. Besides, due to our bit-wise signal embedding approach, the representation also contain some bit-level information of variables, which support operations like bit-and (`&`), bit-or ( `|` ). However, we also find limitations of current representation. The model struggles to predict the exact values of variables. For assertion like `v≠2` and `v≠4`, the prediction accuracy is lower, which leaves room for further improvement.

Additionally, Table 3 also presents prediction performance of DR-GNN$_{vs}$ (trained only with variable trigger supervision) and DR-GNN$_{bs}$ (trained only with branch hit supervision). The original DR-GNN exhibits best performance, which demonstrates the benefits of both supervisions. It also proves that supplementing more high-quality supervisions could help us learn a more comprehensive representation of circuit dynamic behaviors.

# 6 RELATED WORK

**Learning representation for circuit:** Circuits can be represented as graphs, with operators as nodes and wires as edges, making GNNs useful for learning circuit structure representations. For instance, TAG (Zhu et al., 2022) use GNNs for analog and mixed-signal circuit representation, aiding in layout matching prediction and wirelength estimation. ABGNN (He et al., 2021) is used for digital circuit representation and arithmetic block identification.

In addition to circuit structure learning, recent studies have increasingly focused on learning functionality-aware circuit representations. For instance, FGNN (Wang et al., 2022) uses contrastive learning to differentiate between functionally equivalent and inequivalent circuits. DeepGate family Li et al. (2022); Shi et al. (2023; 2024) is also a pioneering approach, which transforms circuit netlists into And-Inverter Graphs (AIGs) and uses node logic-1 probability or truth table similarity as training supervision. However, these methods can only handle combinational circuits, falling short in representing sequential circuits, which are more common in the real world. Moreover, although these representations in netlist level are useful in tasks like logic synthesis and PPA prediction, they make little contribution in the front end of circuit design, which needs a representation in RTL level. Fang et al. (2023) tries to construct a pre-synthesis PPA estimation framework, but its representation is between netlist level and RTL level, not an authentic RTL format.

Our work focus on learning dynamic representations of RTL hardware designs with sequential circuit behaviors. The most similar work to ours is Design2Vec (Vasudevan et al., 2021), which also attempts to predict the status of cover point hits under different test case parameters. However, Design2Vec does not learn a universal representation for circuits and trains a distinct GNN model for each design. Besides, the statement-level CDFG used in Design2Vec is code-oriented, where each node in the graph represents a statement. While this approach effectively captures RTL code semantics, it cannot clearly reflects dynamic circuit data flow. DeepSeq (Khan et al., 2024) is another attempt learning sequential circuits. It focues on AIG at netlist level and predicts the logic-1 and state transition probability under random workload. Without considering the different circuit behaviors under different inputs, its representation significantly lacks dynamic information compared to ours. Meanwhile, our work learns circuits in RTL level, providing a more high-level graph compared to netlist, which makes our model more suitable for dynamic representation.

**Learning representation for software program:** Recent studies have increasingly applied machine learning to software code representation (Allamanis et al., 2018). A subset of these studies also focus on understanding dynamic behavior of programs, a concept known as *learning to execute*. Zaremba & Sutskever (2014) firstly uses RNN to read the program character-by-character and computes the program's output. Bieber et al. (2020) transforms programs into statement-level control flow graphs, employing a custom IPA-GNN for output prediction. Shi et al. (2019) focuses on assembly code, converting it into instruction-level control flow graphs and using GNN to predict the next execution branch and prefetch address.

Compared to hardware description code, software programs predominantly follow a serial execution logic, executing one statement at a time. Differently, hardware code represents concurrent data flows between hardware components, signifying a fundamental divergence from software code. Furthermore, the same circuit code is executed in each clock cycle, leading to dynamic behaviors (*i.e.,* state transition space) that could have a much deeper depth than software. This makes learning dynamic hardware representation more challenging than software.

## 7 CONCLUSION

We present an approach that is able to learn representations for digital circuits to predict their dynamic behaviors. For the first time, we demonstrate the ability of deep learning in capturing the complex temporal logic in sequential circuits. We learn the RTL representations which can generalize across different designs and different tasks. Since DynamicRTL learns over the RTL CDFG, it can also potentially generalize to other dynamic tasks in RTL level such as debugging, model checking and test case generation. More broadly, this work shows the power and potential of deep learning to effectively integrate with hardware domain, which facilitates a high-level understanding of circuits and promotes more efficient circuit design processes.

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

# A  MODEL

## A.1  PROJECT LINK

The source code of DynamicRTL and our collected dynamic circuit dataset is available at the anonymous link `https://anonymous.4open.science/r/DynamicRTL-FA33`.

## A.2  HYPERPARAMETERS

The hyperparameters for the DR-GNN model are given in Table 4.

Table 4: Hyperparameters for models.

| | | |
|---|---|---|
| DR-GNN | input feature size | 1536 |
| | operation embedding size | 32 |
| | position embedding size | 32 |
| | hidden size | 1536 |
| | GNN layer | 20 |
| | GRU layer | 1 |
| | optimizer | adam |
| | learning rate | 0.0001 |
| seq2vec | input number size | 32 |
| | hidden size | 512 |
| | GRU layer | 3 |
| | optimizer | adam |
| | learning rate | 0.0001 |
| MLP for task prediction | layer | 2 |
| | hidden size | 50 |
| | loss | binary cross entropy |
| | learning rate | 0.0001 |

# B    CDFG OF HARDWARE DESIGNS

## B.1    MORE CDFG EXAMPLES

Figure 5, 6 and 7 illustrate more examples of our operation-level circuit control data flow graph.

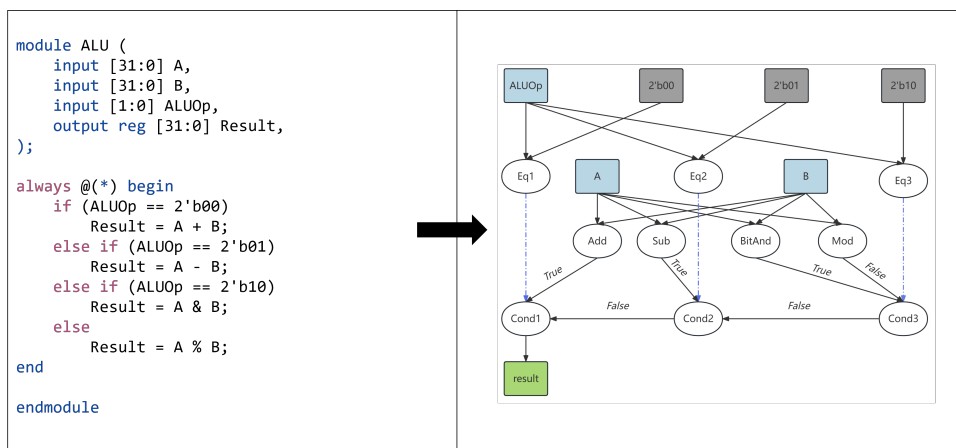

Figure 5: Arithmetic logic unit.

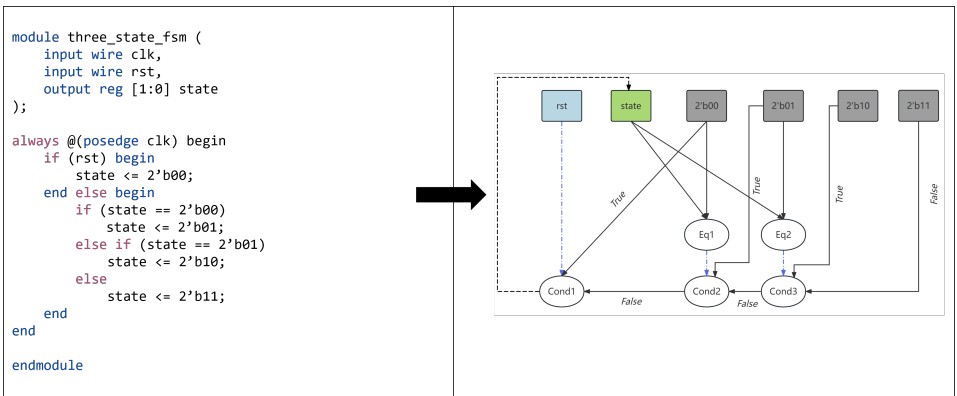

Figure 6: Three-state finite state machine.

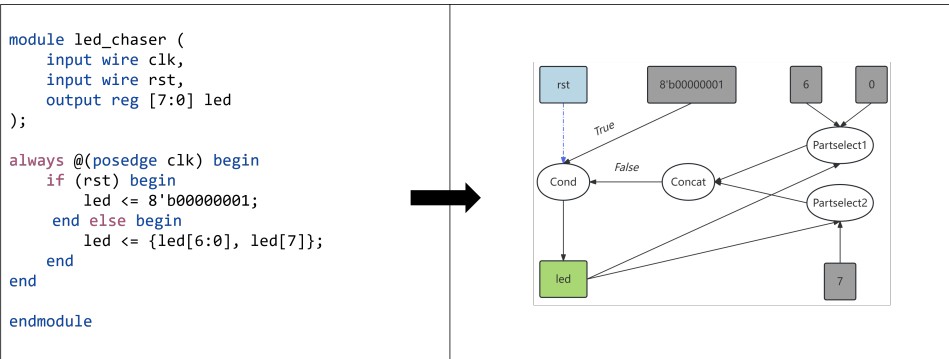

Figure 7: LED chaser.

## B.2  CDFG NODE SUB-TYPES

We describe the CDFG node sub-types in this subsection. Table 5 outlines the basic node types in CDFG. Table 6 provides a detailed introduction of Verilog unary operators and binary operators, which are shortly mentioned in Table 5.

Table 5: Basic node type in circuit CDFG.

| Node-Type | In-Edge | Example | Description |
|---|---|---|---|
| Cond | Sel, Data1, Data2 | `a = b?c:d` | Conditional selection between two data according to the select signal.
*Example : If b is equal to True, assign c to a, otherwise assign d to a.* |
| PartSelect | Var, High, Low | `a = b[x:y]` | Select a part of a variable from the high index to low index.
*Example: Assign bits from position x to y of b to a.* |
| Concat | Src1, Src2, ... | `a = {b,c,d}` | Concatenate multiple variables together.
*Example : Connect b, c, and d together in order and assign them to a.* |
| Unary Operator | Src | `a = op1 b` | Unary operator with single source operand. |
| Binary Operator | Src1, Src2 | `a = b op2 c` | Binary operator with two source operands. |
| Register | - | `reg [7:0] a` | Register type variable in Verilog.
*Example : An 8-bit register type variable named 'a'.* |
| Wire | - | `wire [7:0] b` | Wire type variable in Verilog.
*Example : An 8-bit wire type variable named 'b'.* |
| Constant | - | `32'd7` | The constant in Verilog.
*Example : A 32-bit constant representing decimal value 7.* |

Table 6: Detailed node types of unary and binary operators in circuit CDFG.

| Operator-Type | Example | Description |
|:---:|:---:|:---:|
| LNot | `!a` | Logical Not |
| Not | `~a` | Bitwise Not |
| Lt | `a < b` | Less than |
| Le | `a <= b` | Less than or equal to |
| Gt | `a > b` | Greater than |
| Ge | `a >= b` | Greater than or equal to |
| Add | `b + c` | - |
| Sub | `b - c` | - |
| Mul | `b * c` | - |
| Div | `b / c` | - |
| Mod | `b % c` | - |
| ShiftLeft | `b << c` | Logical left shift |
| ShiftRight | `b >> c` | Logical right shift |
| AshiftLeft | `b <<< c` | Arithmetic left shift |
| AshiftRight | `b >>> c` | Arithmetic right shift |
| And | `b && c` | - |
| Or | `b || c` | - |
| Eq | `b == c` | - |
| Neq | `b != c` | - |
| BitAnd | `b & c` | Bitwise And |
| BitOr | `b | c` | Bitwise Or |
| BitXor | `b ^ c` | Bitwise Xor |
| URxor | `^a` | Xor on each bit |
| URand | `&a` | And on each bit |
| URor | `|a` | Or on each bit |

# C    CIRCUIT DATASET DETAILS

This section provides a supplementary introduction to our circuit dataset. We define circuit size based on the number of nodes in the circuit control data flow graph. In our dataset, over 95% of the circuits contains fewer than 400 CDFG nodes. Figure 8 illustrates the distribution of circuit sizes. Table 7 illustrates the actual scale of circuits, featuring Verilog files with up to 1,000 lines and netlists containing up to 50,000 nodes.

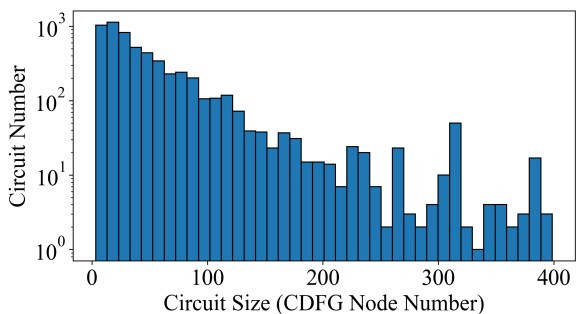

Figure 8: Histogram of circuit size (CDFG node number) in our dataset.

| Range | Count | | Range | Count | | Range | Count |
|---|---|---|---|---|---|---|---|
| | | | | | | (0, 200] | 2469 |
| (0, 100] | 3172 | | (0, 100] | 3048 | | (200, 500] | 1371 |
| (100, 200] | 1458 | | (100, 200] | 1517 | | (500, 1000] | 932 |
| (200, 500] | 1023 | | (200, 500] | 1002 | | (1000, 2000] | 653 |
| (500, 1000] | 351 | | (500, 1000] | 389 | | (2000, 5000] | 450 |
| 1000+ | 320 | | 1000+ | 368 | | (5000, 10000] | 125 |
| | | | | | | (10000, 50000] | 142 |
| (a) Lines of Verilog | | | (b) Lines of Verilog IR | | | 50000+ | 182 |
| | | | | | | (c) Nodes of Aiger | |

Table 7: Summary of Circuit Scale

To ensure the diversity of our circuit dataset and eliminate potential biases towards specific circuit types, we employed t-Distributed Stochastic Neighbor Embedding (t-SNE) for visualization. The circuit feature consists of the counts of each type of nodes present in the circuit. The resulting mapping, as illustrated in the Figure 9, shows that most circuits exhibit an average distribution, thereby demonstrating the diversity of our dataset.

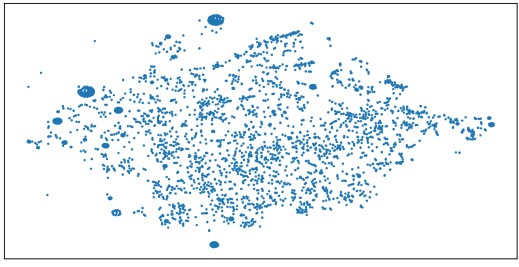

Figure 9: T-SNE Visualization of Circuit Dataset

