# OpenReview forum: "DynamicRTL: RTL Representation Learning for Dynamic Circuit Behavior"
_ICLR.cc/2025/Conference — Submitted to ICLR 2025_

### Official Review · Reviewer_aUtV · 2024-10-27

**Soundness:** 2
**Presentation:** 2
**Contribution:** 2
**Rating:** 6
**Confidence:** 3

**Summary:**

The DynamicRTL paper introduces a Graph Neural Network model that captures dynamic behaviors in RTL circuits. Unlike traditional models that focus on static structures, DynamicRTL uses Control Data Flow Graphs (CDFGs) to incorporate both static and dynamic data, enabling more accurate branch execution and assertion predictions. The model achieves good accuracy in branch prediction and demonstrates strong transfer learning potential for tasks like assertion prediction. DynamicRTL offers insights for hardware verification and optimization, setting a new standard for understanding dynamic circuit behaviors.

**Strengths:**

The DynamicRTL paper introduces an approach that combines static and dynamic circuit information. The model delivers results that outperforming other methods in tasks like branch prediction and assertion prediction, while also contributing a dynamic circuit dataset. The paper is clear in its presentation, offering explanation of the CDFG construction and the DR-GNN model’s architecture. Lastly, DynamicRTL tackles a significant problems for the evolving field of hardware design and electronic design automation (EDA).

**Weaknesses:**

The weaknesses of the DynamicRTL paper are:
it relies heavily on existing GNN methods and concepts like CDFGs, making it less groundbreaking in terms of underlying technology.
The model’s generalizability to other circuit types or levels, such as netlists, remains unexplored, and its performance on larger or more complex circuits could be tested more rigorously.
Additionally, the paper’s significance is limited, as it primarily targets RTL-level circuits, restricting its applicability to broader hardware contexts or higher abstraction levels, such as layout optimization tasks.

**Questions:**

The paper would benefit from including a brief mention of the results in the abstract to give readers a clearer overview of its performance. Additionally, it would be useful to know how the proposed design compares with existing commercial tools in terms of efficiency and accuracy. Finally, more details on how input sequences are selected for testing would be helpful to understand the robustness of the model in different scenarios.

---

> ### Author Response · Authors · 2024-11-18
>
> Although learning representations on CDFGs of programming languages is common, to the best of our knowledge, there is no open-source work capable of building complete operator-level CDFGs of Verilog. This task is more challenging compared to programming languages because bit-level operators are commonly used, which makes the CDFG-building process significantly more complex. Our work overcomes this challenge by building operator-level CDFGs from Verilog and using simulation results generated by a simulator as supervision.
>
> We tested our model at the module level of RTL, but this does not mean that our experiments were only conducted on circuits of small scale. One of our key motivations for acquiring word-level CDFG representations of Verilog is that word-level representations can reduce the graph's scale while preserving the semantic information of the original RTL code. We present the statistics of the lines of original Verilog, Verilog IR, and Aiger (an intermediate representation of netlist-level circuits after decomposing) in the tables below. These statistics reveal that the actual scale of circuits is not small; a portion of the circuits exceeds the representation capability of most current LLMs and GNNs. For instance, an RTL-level circuit graph with 100 nodes can be transformed into an AIG graph with more than 10,000 nodes after synthesis, resulting in the loss of the semantic information of Verilog.
>
> Overall, thank you again for your valuable feedback. We hope that our response clarifies the contributions and novelty of our work and encourages you to reconsider your previous rating of the paper.
>
> **Lines of Verilog**
>
> |  **Range**  | **Count** |
> | :---------: | :-------: |
> |  (0, 100]   |   3172    |
> | (100, 200]  |   1458    |
> | (200, 500]  |   1023    |
> | (500, 1000] |    351    |
> | 1000+ |    320    |
>
>
> **Lines of Verilog IR**
>
> |  **Range**  | **Count** |
> | :---------: | :-------: |
> |  (0, 100]   |   3048    |
> | (100, 200]  |   1517    |
> | (200, 500]  |   1002    |
> | (500, 1000] |    389    |
> | 1000+  |    368    |
>
>
> **Nodes of Aiger**
>
> |   **Range**   | **Count** |
> | :-----------: | :-------: |
> |   (0, 200]    |   2469    |
> |  (200, 500]   |   1371    |
> |  (500, 1000]  |    932    |
> | (1000, 2000]  |    653    |
> | (2000, 5000]  |    450    |
> | (5000, 10000] |    125    |
> | (10000, 50000]  |    142    |
> |  50000+ | 182|

---

> ### Comment · Reviewer_aUtV · 2024-11-25
> **rebuttal received**
>
> Thank you for the reply and clarification. Score raised.

---

> > ### Author Response · Authors · 2024-11-26
> >
> > Thank you very much for your positive feedback and for taking the time to reconsider your evaluation of our paper.

---

### Official Review · Reviewer_df1P · 2024-10-28

**Soundness:** 2
**Presentation:** 3
**Contribution:** 2
**Rating:** 3
**Confidence:** 4

**Summary:**

This work introduces DR-GNN, a GNN-based model for learning circuit representations via incorporating dynamic information. Specifically,  DR-GNN integrates static structures with multi-cycle execution dynamics using a Control Data Flow Graph (CDFG) representation of RTL circuits. Trained on a newly developed dataset of over 6,300 Verilog modules and 190,000 simulation traces, DR-GNN achieves promising results on branch prediction tasks compared to conventional GNN models. Additionally, it demonstrates transferability to related tasks like assertion prediction.

**Strengths:**

- Overall, the paper is well written. All technical steps are easy to follow.
- Learning dynamic circuit representation is both an interesting and important problem.

**Weaknesses:**

- Limited Novelty: As the authors noted, the proposed model architecture is largely based on the Heterogeneous Graph Transformer [1], and the use of GRU has also been explored in numerous prior works (e.g., [2]). Therefore, I regret to say that the novelty of this work is insufficient.
- Missing Stronger and More Relevant Baselines: Several recent circuit representation learning methods (e.g., [3-5]) are omitted from the experiments. Notably, [3] also proposes learning different aggregation functions for different node types. Comparing only with basic GNNs like GCN does not fully convince me of the proposed approach's effectiveness.
- Unconvincing Claims: While the authors claim that DR-GNN overcomes the over-smoothing issue, Figure 4 shows a significant drop in accuracy with 30 layers. Additionally, there are numerous prior works addressing over-smoothing [6], yet they are discussed without comparison in this study.
- Limited Reproducibility: The manuscript lacks detailed hyperparameter settings for DR-GNN and baseline GNNs, making it unclear how hyperparameters were tuned. More importantly, since the source code and generated circuit dataset are not provided, it is nearly impossible to reproduce the results presented in this work.

[1] Hu et al., "Heterogeneous Graph Transformer", WWW'20. \
[2] Li et al., "Gated Graph Sequence Neural Networks", ICLR'16. \
[3] Wang et al., "Functionality Matters in Netlist Representation Learning", DAC'22. \
[4] Yang et al., "Versatile Multi-stage Graph Neural Network for Circuit Representation", NeurIPS'22. \
[5] Deng et al., "Less is More: Hop-Wise Graph Attention for Scalable and Generalizable Learning on Circuits", DAC'24. \
[6] Rusch et al., "A Survey on Oversmoothing in Graph Neural Networks", arXiv'23.

**Questions:**

- It appears the authors split the circuit dataset into only training and test sets, with no validation set. How was hyperparameter tuning performed in this case?
- What is the variance in accuracy scores reported in Tables 1 and 3?
- How does the model’s accuracy vary across circuit designs with different depths (in contrast to different sizes in Table 2)?

---

> ### Author Response · Authors · 2024-11-18
>
> Thank you for your valuable review.
> ### Weakness1: The proposed model architecture is largely based on the Heterogeneous Graph Transformer.
> R: We acknowledge that we use the basic structure of HGT and GRU, which are common techniques in ML. The main contribution of our work lies in transferring these models into the circuit domain, which is a novel practice for this field. **The difficulty lies in how to map a circuit into a heterogeneous graph.** To address this, we designed the CDFG data structure to complete this task. We introduced novel designs for position embedding and operation embedding, and explored the embedding pattern of circuit sequence data. These designs are all tailored to circuit graph learning, and we believe they will be helpful for future hardware researchers. Specifically, **we implement position embedding to help the model recognize the order of operands during message passing, as some operators do not satisfy the commutative property**.
>
> ### Weakness2: Missing Stronger and More Relevant Baselines
> R: We should clarify that the research you referred to focuses on different tasks and circuit levels compared to our work. These works primarily address prediction tasks like PPA prediction, which are based on circuit graphs closer to the netlist (e.g., And-Inverter Graphs). However, our work focuses on learning circuit functionality at the RTL level and performing tasks like branch prediction and assertion prediction. Using models designed for netlist learning in RTL-level tasks is extremely challenging. For instance, branches in RTL-level circuits are difficult to map to wires in circuit netlists. Additionally, the graph scale differs significantly. For example, an RTL-level circuit graph with 100 nodes can be transformed into an AIG graph with more than 10,000 nodes after synthesis. It is not feasible to use such models to effectively learn circuit functions at the RTL level. Furthermore, we compare our model against Design2Vec [1], which focuses on the same task as ours and is trained on the same dataset. Experiments show that our model significantly outperforms Design2Vec.
>
> ### Weakness3: Over-smoothing issue
> R: Thank you for your valuable advice. We will explore the techniques referenced in your work and incorporate them to further improve our model's performance.
>
> ### Weakness4: The source code and generated circuit dataset are not provided
> R: **We have open-sourced our framework and dataset** at https://anonymous.4open.science/r/DynamicRTL-FA33, as shown in the Appendix. We have also provided the model hyperparameters in the Appendix. We believe this will help hardware researchers reproduce our work.
>
> ### Q1: The split of train, test, val dataset. The hyperparameter tuning performed in this case.
> R: There are several key parameters that greatly influence the model's performance, such as the GNN layer and the GRU layer in the GNN. During the training stage, we observe the accuracy of the pretraining task (branch prediction) in the first few epochs under different hyperparameter settings. We then use the best parameters to complete training and perform evaluation.
>
> ### Q2: What is the variance in accuracy scores reported in Tables 1 and 3?
> R: Table 1 and Table 3 correspond to different tasks. Table 1 shows the model's performance on the branch prediction task, while Table 3 evaluates the model pretrained on the branch prediction task for the assertion prediction task. These experiments demonstrate that our model learns dynamic information about circuits and can be transferred to other tasks related to the dynamic functionality of circuits.
>
> ### Q3: The model’s accuracy vary across circuit designs with different depths?
> R: The average depth of circuit CDFGs in our dataset is 7, and the depths of most circuit CDFGs are under 10 (as noted in line 372 of the paper). This low depth is due to the characteristics of circuits. If a circuit's depth is too high, its latency becomes very large, degrading its performance. Consequently, circuits are typically implemented with parallel structures, which is a key difference between Verilog and other programming languages. For your third question, we have supplemented the experimental results as the table shown below.
>
> Overall, thank you again for your valuable feedback. We hope that our response clarifies the contributions and novelty of our work and encourages you to reconsider your previous rating of the paper.
>
> |   **Depth Range**   | **Branch Hit** | **Variable Trigger** |
> | :-----------: | :-------: | :-------: |
> |   (0, 5]   | 98.27 | 92.93 |
> |   (5, 10] | 93.16 | 89.15 |
> |   10+ | 89.74 | 85.46 |
>
> [1] Vasudevan et al. "Learning semantic representations to verify hardware designs". NIPS 2021

---

> ### Author Response · Authors · 2024-11-26
> **Size of Circuits**
>
> **Lines of Verilog**
> |  **Range**  | **Count** |
> | :---------: | :-------: |
> |  (0, 100]   |   3172    |
> | (100, 200]  |   1458    |
> | (200, 500]  |   1023    |
> | (500, 1000] |    351    |
> | 1000+ |    320    |
>
>
> **Lines of Verilog IR**
> |  **Range**  | **Count** |
> | :---------: | :-------: |
> |  (0, 100]   |   3048    |
> | (100, 200]  |   1517    |
> | (200, 500]  |   1002    |
> | (500, 1000] |    389    |
> | 1000+  |    368    |
>
>
> **Nodes of Aiger**
> |   **Range**   | **Count** |
> | :-----------: | :-------: |
> |   (0, 200]    |   2469    |
> |  (200, 500]   |   1371    |
> |  (500, 1000]  |    932    |
> | (1000, 2000]  |    653    |
> | (2000, 5000]  |    450    |
> | (5000, 10000] |    125    |
> | (10000, 50000]  |    142    |
> |  50000+ | 182 |

---

> ### Comment · Reviewer_df1P · 2024-11-28
> **Follow-up**
>
> Thanks authors' response. My remaining concerns are as follows:
>
> >The difficulty lies in how to map a circuit into a heterogeneous graph. To address this, we designed the CDFG data structure to complete this task.
>
> The conversion of RTL to the CDFG data structure appears primarily as an engineering step. At line 132 of the manuscript, the authors themselves even note that utilizing a CDFG is **a common way** to understand the function of RTL. Thus, the conversion step may not represent a novel advancement but rather a standard practice within the field
>
> >we implement position embedding to help the model recognize the order of operands during message passing, as some operators do not satisfy the commutative property.
>
> First of all, the authors do not mention position embeddings in the list of contributions in the Introduction section, leading readers to believe that this is not a key contribution of this work. Moreover, there are numerous positional embedding methods (some detailed in GraphGPS[7]), none of which are compared against in this study. I believe a more rigorous empirical comparison would make the efficacy of the proposed method more convincing.
>
> >These works primarily address prediction tasks like PPA prediction, which are based on circuit graphs closer to the netlist...Using models designed for netlist learning in RTL-level tasks is extremely challenging.
>
> While the referenced methods focus on GNNs specifically designed for netlists, it's unclear why the authors have not considered applying these models directly to the CDFG generated from RTL in this study. For example, I see no apparent technical barrier that would prevent replacing the aggregation step in this paper with the aggregation functions proposed in [3].
>
> >Over-smoothing issue
>
> It seems authors haven't resolved this concern yet. The current manuscript still mentions the proposes approach can mitigate GNN over-smoothing, while not being supported by empirical results.
>
> >The split of train, test, val dataset. The hyperparameter tuning performed in this case.
>
> I would like to clarify my previous question. Since authors split the circuit dataset into only training and test sets, with no validation set, do authors only use training set to perform hyperparameter tuning? This experimental setup is unclear from either the manuscript or authors' response.
>
> Given all the concerns mentioned above, I have decided to maintain my current score.
>
> [7] Rampášek et al., "Recipe for a General, Powerful, Scalable Graph Transformer", NeurIPS'22.

---

### Official Review · Reviewer_wxWe · 2024-11-02

**Soundness:** 2
**Presentation:** 3
**Contribution:** 2
**Rating:** 5
**Confidence:** 4

**Summary:**

The authors propose a representation learning approach for RTL descriptions of circuits. Unlike prior works that generally learns the static structure of the circuits, this work is capable to cover the dynamic execution features of circuits which change with inputs and circuit states. Specifically, as  the circuit’s functionality is embedded in the attributes of the nodes and their connectivity, the authors model the RTL as an operational-level CDFG and apply GNNs to learn circuit representations. Notably, because some operator nodes in circuits are non-commutative (e.g., subtraction), the authors parameterize the weights of the heterogeneous attention based on different operation types and node positions, enabling effective aggregation of different circuit operations. Finally, the authors verify the proposed GNN model with two downstream tasks.

**Strengths:**

- This work targets at representation of circuits with dynamic behaviors for the first time.

- This work addresses the over-smoothing problem encountered in the GNN model.

- This work also builds a benchmark for dynamic circuit modeling.

**Weaknesses:**

1) The dynamic characters of the circuit mentioned in this work mainly include branch and multi-cycle execution. Do they cover all the dynamic features appropriately? It needs to be defined more clearly and formally. For instance, there are many iterative algorithms such as equation solvers implemented with hardware, but the number of iterations depends on the amount of the error and it affects the runtime and power consumption of the circuit substantially. Can it be estimated given the initial inputs using the proposed GNN model?

2) The experiments are relatively weak. The basic goal of representation of circuits with dynamic characteristics is PPA estimation, but it is not mentioned nor compared to prior works. The proposed downstream tasks are not quite typical tasks. In addition, the circuits used by the authors (with node counts ranging from 10 to 500) are too small which is insufficient to validate the proposed approach.

**Questions:**

See the weakness.

---

> ### Author Response · Authors · 2024-11-18
>
> Thank you for your review.
>
> For the first question, we acknowledge that our model cannot handle all the dynamic features of circuits. We focus on a specific area that has been neglected by previous research: the dynamic functionality at the RTL level. Compared with other dynamic features, such as PPA, RTL-level functionality prediction requires learning a more fine-grained representation of circuits with simulation data. As for the iterative algorithms you mentioned, they indeed present a good scenario for GNN models to improve upon. We believe that with sufficient specific training data, our model could address this question, and this could be considered one of our future directions.
>
> For the second question, there are numerous works that focus on predicting the PPA of circuits, and most of them predict PPA at the netlist level, which is closer to the implementation and actual scale of circuits. What sets our work apart is that it focuses on the dynamic behavior representation of circuits at the RTL level, which is closer to the semantic information and functionality of circuits and is more readable for humans. If someone wants to predict the PPA of circuits, they can synthesize the RTL circuits into netlists to obtain more precise PPA values. Moreover, the process of synthesis is not time-consuming.
>
> When it comes to the scale of circuits, the motivation for acquiring the word-level CDFG representation of Verilog lies in its ability to downsize the scale of the graph while preserving the semantic information of the original RTL code. We present the statistics of the lines of original Verilog, Verilog IR, and nodes of Aiger (an intermediate representation of a netlist-level circuit after decomposing the circuits) below. These statistics reveal that the actual scale of circuits is not small; a portion of the circuits exceeds the representation capability of most current LLMs and circuit learning models. This highlights a significant contribution of our paper.
>
> Overall, we sincerely thank you again for your valuable feedback. We hope that our response clarifies the contributions and novelty of our work, encouraging you to reconsider your previous rating of the paper.
>
> **Lines of Verilog**
> |  **Range**  | **Count** |
> | :---------: | :-------: |
> |  (0, 100]   |   3172    |
> | (100, 200]  |   1458    |
> | (200, 500]  |   1023    |
> | (500, 1000] |    351    |
> | 1000+ |    320    |
>
>
> **Lines of Verilog IR**
> |  **Range**  | **Count** |
> | :---------: | :-------: |
> |  (0, 100]   |   3048    |
> | (100, 200]  |   1517    |
> | (200, 500]  |   1002    |
> | (500, 1000] |    389    |
> | 1000+  |    368    |
>
>
> **Nodes of Aiger**
> |   **Range**   | **Count** |
> | :-----------: | :-------: |
> |   (0, 200]    |   2469    |
> |  (200, 500]   |   1371    |
> |  (500, 1000]  |    932    |
> | (1000, 2000]  |    653    |
> | (2000, 5000]  |    450    |
> | (5000, 10000] |    125    |
> | (10000, 50000]  |    142    |
> |  50000+ | 182 |

---

> ### Comment · Reviewer_wxWe · 2024-11-24
>
> Thank you for the replies, but some of the replies are still not quite clear.
> 1) If the design can also handle iterative designs, it will be good to take iterative designs such as graph processing accelerators into consideration.
>
> 2) "What sets our work apart is that it focuses on the dynamic behavior representation of circuits at the RTL level, which is closer to the semantic information and functionality of circuits and is more readable for humans.". RTL-level representation has better semantic advantages, but specific benefits more than than "readable for humans" are expected. By the way, PPA indicates power, performance, and area. I agree that power and area are usually good enough using logic synthesis, but performance is not supported in most of the synthesis tools. In fact, if the dynamic features such as branch and iterative execution are sufficiently considered, it will produce better performance prediction and can be compared to netlist-based prediction.
>
> 3) The benchmark looks reasonable for me. I think the setups in the experiment section is bit misleading which can be improved later.

---

> ### Author Response · Authors · 2024-11-26
> **Response to Reviewer wxWe 2**
>
> Thanks a lot for your valuable advice.
>
> ### Q1 & Q2: It will be good to take iterative designs such as graph processing accelerators into consideration.
>
> R: Maybe I start to understand the meaning of your advice after thinking about it. The **advice 1** meanings that we can take algorithm circuits implemented in hardware into consideration. And possibly, the meaning of **advice 2** is to consider the potential of evaluating the performance of the algorithm circuits implemented in hardware, not the delay of logic gates. Because if the meaning of “performance” there is the delay of logic gates in circuits, it still can be derived easily by netlist-level representation. May I ask if my understanding is correct?
>
> We haven’t explored the potential of our work on algorithm circuits implemented in hardware, as our initial goal was to derive representations for general circuits. To achieve this, we collected data from the Internet and ensured that the data distribution was balanced. Considering this focus, we do not have much experience with algorithm circuits implemented in hardware. These circuits are somewhat niche and are often implemented using FPGA. However, we plan to expand our work into this field in the future.
>
> ### Q3:  The benchmark looks reasonable. I think the setups in the experiment section is bit misleading which can be improved later.
>
> R: Thanks a lot for your advice, we have revised it.

---

### Official Review · Reviewer_q4C3 · 2024-11-03

**Soundness:** 3
**Presentation:** 3
**Contribution:** 3
**Rating:** 6
**Confidence:** 4

**Summary:**

The paper focuses on improving upon the current state of GNNs for circuit design that depend on the static data of circuits by including dynamic behaviors. The paper creates a large dataset with various designs and simulation traces, then build a dynamic RTL GNN model that is built on the circuit operation-level graph. The paper shows the effectiveness of this representation on branch prediction, then shows that this can be transferred to assertion prediction task.

The paper seems to improve by exploring an underexplored dimension of dynamism in circuit-GNNs. The paper is well organized and shows that it can perform well on a given task + transferred to a downstream task that hints the effectiveness of this new representation and of adding dynamism to the circuit-GNNs.

**Strengths:**

The paper seems to improve by exploring an underexplored dimension of dynamism in circuit-GNNs. The paper is well organized and shows that it can perform well on a given task + transferred to a downstream task that hints the effectiveness of this new representation and of adding dynamism to the circuit-GNNs.

**Weaknesses:**

The task that was used seems to be too simple to say that it is effective. Seems like the tasks are simply binary classification tasks which is not sufficient.

**Questions:**

* Can you provide more details about the test set for the two tasks? For example, size of the dataset, is it balanced, ... Without that information, it is possible for the data to be biased.

* It seems that the number of nodes in the circuit are too small. How does this scale for much larger ones? At what point does the accuracy drop significantly for table 2.

* How is the speed of inference and training?

* For the two tasks, how does it compare to human evaluation or any pre-existing programs?

---

> ### Author Response · Authors · 2024-11-18
>
> Thank you for your review.
> ### Weakness: The task that was used seems to be too simple to say that it is effective. Seems like the tasks are simply binary classification tasks which is not sufficient.
> R: Branch prediction accuracy is a commonly used metric, as seen in Design2Vec [1]. Although it is simply a binary classification task, accurately predicting it is quite challenging because it is strongly related to the input sequence and fine-grained circuit functionalities. We have also tried to enrich the metrics in our paper, this is why we introduced the assertion prediction metrics, which further demonstrate that we have learned a comprehensive representation of circuit dynamic behaviors using just branch prediction tasks. **Branch coverage prediction and assertion are both important tasks in the ASIC verification flow**, which accounts for about 40% of the time spent on IC design and tape-out.
>
> ### Q1: Can you provide more details about the test set for the two tasks? For example, size of the dataset, is it balanced, ... Without that information, it is possible for the data to be biased.
> R1: We have provided **details about our dataset in Appendix C**. To show that our dataset is unbiased, we applied the t-SNE method, and the results are also included in the Appendix.
>
> ### Q2: It seems that the number of nodes in the circuit are too small. How does this scale for much larger ones? At what point does the accuracy drop significantly for table 2.
> R2: One of our key motivations for acquiring the word-level CDFG representation of Verilog is that word-level representation can reduce the graph's scale while preserving the semantic information of the original RTL code. **We present statistics on the lines of original Verilog, Verilog IR, and nodes of Aiger (an intermediate representation of netlist-level circuits after decomposing the circuits) in the tables below.** These statistics reveal that the actual scale of circuits is not small; a portion of the circuits exceeds the representation capability of most current LLMs and GNNs. For example, an RTL-level circuit graph with 100 nodes can be transformed into an AIG graph with more than 10,000 nodes after synthesis, leading to the loss of the semantic information of Verilog.
>
> ### Q3: How is the speed of inference and training?
> R3: Our model requires approximately 1 hour of training time per epoch on a single A800 GPU for the training dataset. The inference time for each batch is about 0.9 seconds, with a batch size of 128. It is worth mentioning that although the number of our CDFGs is about 6,300, each CDFG includes 30 simulation traces, resulting in approximately 190,000 data entries.
>
> ### O4: For the two tasks, how does it compare to human evaluation or any pre-existing programs?
> R4: Our model requires significantly less time compared to commercial hardware tools, such as Synopsys VCS, to obtain branch prediction results. The key advantage of using our model for branch prediction is that we can compute the branch hit results for a large number of input sequences simultaneously using a GPU, which is much faster than human evaluation or hardware simulators. This is particularly useful in hardware verification scenarios, where fast but less accurate screening of all generated test cases is required.
>
> Overall, thank you again for your review.
>
> **Lines of Verilog**
>
> |  **Range**  | **Count** |
> | :---------: | :-------: |
> |  (0, 100]   |   3172    |
> | (100, 200]  |   1458    |
> | (200, 500]  |   1023    |
> | (500, 1000] |    351    |
> | 1000+ |    320    |
>
> **Lines of Verilog IR**
>
> |  **Range**  | **Count** |
> | :---------: | :-------: |
> |  (0, 100]   |   3048    |
> | (100, 200]  |   1517    |
> | (200, 500]  |   1002    |
> | (500, 1000] |    389    |
> | 1000+  |    368    |
>
> **Nodes of Aiger**
>
> |   **Range**   | **Count** |
> | :-----------: | :-------: |
> |   (0, 200]    |   2469    |
> |  (200, 500]   |   1371    |
> |  (500, 1000]  |    932    |
> | (1000, 2000]  |    653    |
> | (2000, 5000]  |    450    |
> | (5000, 10000] |    125    |
> | (10000, 50000]  |    142    |
> |  50000+ | 182|
>
> [1] Vasudevan et al. "Learning semantic representations to verify hardware designs". NIPS 2021

---

### Official Review · Reviewer_KM46 · 2024-11-04

**Soundness:** 3
**Presentation:** 2
**Contribution:** 3
**Rating:** 5
**Confidence:** 3

**Summary:**

This paper introduces DynamicRTL, a novel approach for learning representations of digital circuit behavior that incorporates both static structure and dynamic (runtime) execution characteristics. The proposed work has the following contributions:
1. First Comprehensive Dynamic Circuit Dataset
2. Novel Graph Neural Network Architecture (DR-GNN)
3. Extensive experiments show the effectiveness of proposed method

Overall, this work represents a significant step forward in applying machine learning to hardware design and verification, particularly in understanding how circuits behave during execution rather than just their static characteristics. This could have important applications in hardware verification, optimization, and electronic design automation.

**Strengths:**

1. This is the first work to create a comprehensive dataset for dynamic circuit behavior.
2. Extensive experiments demonstrate the effectiveness of proposed method.
3. Novel technical approach combining operation-level CDFG with circuit-aware GNN propagation

**Weaknesses:**

1.  The focuses of proposed work primarily on branch prediction as the main indicator of dynamic behavior. How about other aspects of circuit dynamics such as timing, power consumption, or glitch behavior.
2. It will be better if the authors have some discussion on the limitation of proposed method, especially the effectiveness and efficiency of proposed method for large-scale circuits.

**Questions:**

Please check weakness.

---

> ### Author Response · Authors · 2024-11-18
>
> Thank you for your review.
> ### Weakness1: The focuses of proposed work primarily on branch prediction as the main indicator of dynamic behavior. How about other aspects of circuit dynamics such as timing, power consumption, or glitch behavior.
> R1: We acknowledge that there have been numerous works focusing on PPA prediction, and many of these works have already achieved very good performance. Therefore, we do not aim to further improve their methods in our work. Instead, we focus on other dynamic tasks that have been neglected by previous methods—branch prediction and assertion prediction. Note that our dynamic tasks are quite different from PPA prediction. PPA prediction usually involves learning the global dynamic behavior of circuits, whereas branch prediction focuses on fine-grained dynamic behavior. Most previous PPA prediction works are conducted at the netlist level of circuits, which are closer to the implementation and factual scale of circuits. In contrast, our work focuses on the dynamic behavior representation of circuits at the RTL level, which is closer to the semantic information and functionality of circuits, making it more readable for humans. The significance of our work lies in its focus on dynamic RTL-level circuit representation, which has the potential to enable digital circuit verification. This allows RTL coders to check whether the functionality and semantic information of the RTL satisfy the constraints they set.
>
> ### Weakness2: It will be better if the authors have some discussion on the limitation of proposed method, especially the effectiveness and efficiency of proposed method for large-scale circuits.
> R2: We did test our model at the module level of RTL, but this does not mean that our experiments were only conducted on circuits of small scale. Our motivation for acquiring the word-level CDFG representation of Verilog lies in its ability to downsize the graph's scale while preserving the semantic information of the original RTL code. We present the statistics of the lines of original Verilog, Verilog IR, and Aiger (an intermediate representation of netlist-level circuits) after decomposing the circuits in the tables below. These statistics reveal that the actual scale of circuits is not small; a portion of the circuits exceeds the representation capability of most current LLMs and GNNs. For example, an RTL-level circuit graph with 100 nodes can be transformed into an AIG graph with more than 10,000 nodes after synthesis, leading to the loss of the semantic information of Verilog.
>
> One **limitation** of our work is the complexity of building CDFGs from Verilog. Given the lack of existing work on Verilog CDFG construction, we have not yet attempted to construct repository-level CDFGs but have instead focused on module-level Verilog. Nevertheless, repository-level Verilog CDFG construction is feasible. The original motivation for our work stems from the scarcity of Verilog source code available on the Internet, which significantly weakens the Verilog generation capabilities of LLMs compared to mainstream programming languages. The main purpose of designing digital circuits is computation. Although LLMs can generate Verilog through next-token prediction, they have little understanding of control flow, data flow, and computation. Additionally, the lack of quantity and diversity in Verilog source code limits their ability to generate more correct, creative, and practical Verilog code. Consequently, we believe that CDFG representation learning of Verilog is significant for circuit design automation.
>
> Overall, thank you again for your valuable feedback. We hope that our response clarifies the contributions and novelty of our work and encourages you to reconsider your previous rating of the paper.
>
> **Lines of Verilog**
>
> |  **Range**  | **Count** |
> | :---------: | :-------: |
> |  (0, 100]   |   3172    |
> | (100, 200]  |   1458    |
> | (200, 500]  |   1023    |
> | (500, 1000] |    351    |
> | 1000+ |    320    |
>
> **Lines of Verilog IR**
>
> |  **Range**  | **Count** |
> | :---------: | :-------: |
> |  (0, 100]   |   3048    |
> | (100, 200]  |   1517    |
> | (200, 500]  |   1002    |
> | (500, 1000] |    389    |
> | 1000+  |    368    |
>
> **Nodes of Aiger**
>
> |   **Range**   | **Count** |
> | :-----------: | :-------: |
> |   (0, 200]    |   2469    |
> |  (200, 500]   |   1371    |
> |  (500, 1000]  |    932    |
> | (1000, 2000]  |    653    |
> | (2000, 5000]  |    450    |
> | (5000, 10000] |    125    |
> | (10000, 50000]  |    142    |
> |  50000+ | 182|

---

### Comment · Area_Chair_2KES · 2024-11-23
**Engage in Discussions Before Nov 26 (AoE)**

Dear Reviewers,

First, let me thank you for your invaluable contributions to the ICLR review process. Your constructive feedback plays a key role in enhancing the quality of submissions.

---

As we approach the final days of the discussion phase (ending **Nov 26, 2024, AoE**), I kindly remind you to:

- Please take a moment to review the authors' responses to your comments. This is an opportunity to clarify any remaining questions, acknowledge misunderstandings, and refine your evaluation.

- If you need further clarification, don't hesitate to post your comments as soon as possible.

- If the authors' responses address your concerns or provide new insights, please consider updating your score to reflect this.

---

Your thoughtful participation during this phase is especially valuable for borderline papers, where additional input can be critical to ensuring a fair decision-making process.

I understand how busy this time of year can be and truly appreciate the time and care you dedicate to this important role. Your efforts make a tangible impact on the success of ICLR.

Thank you once again for your dedication.

Best regards,

Area Chair, ICLR 2025

---

### Meta-Review · Area_Chair_2KES · 2024-12-23

**Metareview:**

The paper introduces DynamicRTL, a framework for learning dynamic representations of circuits at the RTL level by combining static structural information with dynamic multi-cycle execution. The authors propose a GNN architecture, which operates on operation-level Control Data Flow Graphs (CDFGs) derived from RTL circuits.

Here are the strengths of the paper:

(a) The dataset is a good contribution to the community (maybe more relevant to dataset tracks), capturing dynamic execution traces and offering potential for future studies in circuit representation learning.

(b) This work addresses the under-explored area of *dynamic* circuit behaviors at the RTL level, providing insights into branch and assertion prediction.

(c) The experiments demonstrate promising results on the proposed tasks, showing transferability of learned representations from branch prediction to assertion prediction.

The reviewers mentioned multiple weaknesses:

(a) The proposed GNN architecture relies on established techniques like Heterogeneous Graph Transformer and GRU, with minimal innovation. The use of operation-level CDFGs is a standard practice in circuit analysis, and while the authors adapt it for dynamic tasks, this seems not to be a significant contribution.

(b) The evaluation focuses only on branch prediction and assertion prediction, which, while relevant, are niche and do not cover broader or more impactful circuit design tasks like timing, power, and area (PPA) estimation.

(c) The baselines are limited to basic GNN models (e.g., GCN, GAT) and an older method (Design2Vec). The paper omits comparisons with more recent state-of-the-art models.

(d) The circuits evaluated are relatively small, with node counts ranging from 10 to 500, leaving scalability to more complex designs untested.

(e) Reviewers raised concerns about hyperparameter tuning, lack of validation datasets, and claims about mitigating over-smoothing in GNNs. The rebuttal responses were insufficiently detailed or dismissive in addressing these points.


While the paper contributes a valuable dataset and targets an interesting problem (dynamic circuit behavior), it lacks the novelty, robustness, and breadth required for acceptance at this stage. The reliance on existing methodologies, narrow task scope, limited scalability, and insufficient comparisons undermine its significance. Additionally, the rebuttal did not satisfactorily address the key concerns raised by reviewers. Therefore, I recommend **Reject** for this work.

**Additional Comments On Reviewer Discussion:**

(Reviewer df1P) argued that the core GNN architecture relies on established techniques like Heterogeneous Graph Transformer and GRU, with minimal innovation. The mapping of RTL to CDFG was criticized as an engineering step rather than a decent contribution. While the reviewer did not engage during rebuttal period, they provided a detailed response to the authors' rebuttal. The authors did not fully address the comments. I understand that the authors could not respond to these additional comments, but after reading them I agree with the reviewer on this and believe that some of the concerns are fundamental to the core idea of the paper.

(Reviewers wxWe, df1P) mentioned that the paper focuses on branch and assertion prediction, which are narrow tasks, and omits comparisons with recent circuit learning methods. The authors defended their choice of tasks, and decided not to perform additional baseline experiments during rebuttal. I also agree with the reviewers on this comment. Without a detailed experimentations on broader range of tasks, it is not clear how valuable such method is.

(Reviewers q4C3, wxWe) The circuits evaluated were relatively small (10–500 nodes), raising questions about scalability to larger designs. The authors argued that CDFG representations reduce graph size while preserving semantic information, and they provided statistics on the dataset to highlight the scale of circuits. However, no experiments were conducted on larger or more complex circuits. I agree that these small experiments are valuable to understand the benefits of the method, however, I would like the authors to study their method on at least one realistic circuit. This would be really useful to show the benefits of their method.

(Reviewers df1P, wxWe) The authors’ claim of mitigating GNN over-smoothing was questioned due to the observed accuracy drop with increasing GNN layers. The authors acknowledged the limitation and suggested exploring referenced techniques in future work. However, no evidence was added to support their mitigation claims.

Given these unresolved concerns and the paper's narrow focus, the recommendation for rejection seems appropriate.

---

### Decision · Program_Chairs · 2025-01-22

Reject